# Discrete Infomax Codes for Supervised Representation Learning

**DOI:** 10.3390/e24040501

**Published:** 2022-04-02

**Authors:** Yoonho Lee, Wonjae Kim, Wonpyo Park, Seungjin Choi

**Affiliations:** 1Stanford AI Lab, Stanford University, Stanford, CA 94305, USA; 2NAVER AI Lab, Seongnam 13561, Korea; wonjae.kim@navercorp.com; 3Standigm, Seoul 06234, Korea; wppark.pio@gmail.com; 4Intellicode & BARO AI Academy, Seoul 06367, Korea; seungjin.choi.mlg@gmail.com

**Keywords:** infomax, discrete codes, representation learning, few-shot classification

## Abstract

For high-dimensional data such as images, learning an encoder that can output a compact yet informative representation is a key task on its own, in addition to facilitating subsequent processing of data. We present a model that produces discrete infomax codes (DIMCO); we train a probabilistic encoder that yields *k*-way *d*-dimensional codes associated with input data. Our model maximizes the mutual information between codes and ground-truth class labels, with a regularization which encourages entries of a codeword to be statistically independent. In this context, we show that the infomax principle also justifies existing loss functions, such as cross-entropy as its special cases. Our analysis also shows that using shorter codes reduces overfitting in the context of few-shot classification, and our various experiments show this implicit task-level regularization effect of DIMCO. Furthermore, we show that the codes learned by DIMCO are efficient in terms of both memory and retrieval time compared to prior methods.

## 1. Introduction

Metric learning and few-shot classification are two problem settings that test a model’s ability to classify data from classes that were unseen during training. Such problems are also commonly interpreted as testing meta-learning ability, since the process of constructing a classifier with examples from new classes can be seen as learning. Many recent works [1,2,3,4] tackled this problem by learning a continuous embedding ($\tilde{x}\in R^{n}$) of datapoints. Such models compare pairs of embeddings using, e.g., Euclidean distance to perform nearest neighbor classification. However, it remains unclear whether such models effectively utilize the entire space of $R^{n}$.

Information theory provides a framework for effectively asking such questions about representation schemes. In particular, the information bottleneck principle [5,6] characterizes the optimality of a representation. This principle states that the optimal representation $\tilde{X}$ is one that maximally compresses the input *X* while also being predictive of labels *Y*. From this viewpoint, we see that the previous methods which map data to $R^{n}$ focus on being predictive of labels *Y* without considering the compression of *X*.

The degree of compression of an embedding is the number of bits it reflects about the original data. Note that for continuous embeddings, each of the *n* numbers in a *n*-dimensional embedding requires 32 bits. It is unlikely that unconstrained optimization of such embeddings use all of these 32n bits effectively. We propose to resolve this limitation by instead using ***discrete embeddings*** and controlling the number of bits in each dimension via hyperparameters. To this end, we propose a model that produces discrete infomax codes (DIMCO) via an end-to-end learnable neural network encoder.

This work’s primary contributions are as follows. We usw mutual information as an objective for learning embeddings, and propose an efficient method of estimating it in the discrete case. We experimentally demonstrate that learned discrete embeddings are more memory and time-efficient compared to continuous embeddings. Our experiments also show that using discrete embeddings help meta-generalization by acting as an information bottleneck. We also provide theoretical support for this connection through an information-theoretic probably approximately correct (PAC) bound that shows the generalization properties of learned discrete codes.

This paper is organized as follows. We propose our model for learning discrete codes in Section 2. We justify our loss function and also provide a generalization bound for our setup in Section 3. We compare our method to related work in Section 4, and present experimental results in Section 5. Finally, we conclude our paper in Section 6.

## 2. Discrete Infomax Codes (Dimco)

We present our model which produces discrete infomax codes (DIMCO). A deep neural network is trained end-to-end to learn *k*-way *d*-dimensional discrete codes that maximally preserve the information on labels. We outline the training procedure in Algorithm 1, and we also illustrate the overall structure in the case of 4-way 3-dimensional codes (k=4,d=3) in Figure 1.
**Algorithm 1** DIMCO training procedure.Initialize network parameters θ**repeat**   Sample a batch $\left\{\left(x^{n},y^{n}\right)\right\}$   Compute logits $\left\{l_{i,j}^{n}\right\}=f\left(x^{n};\theta \right)$            via (Equation 1)   Compute probs $\left\{p_{i,j}^{n}\right\}=\mathrm{softmax}\left(\left\{l_{i,j}^{n}\right\}\right)$     via (Equation 2)   Update θ to minimize loss $L(\left\{p_{i,j}^{n}\right\},\left\{y^{n}\right\})$   via (Equation 10)**until** converged

### 2.1. Learnable Discrete Codes

Suppose that we are given a set of labeled examples, which are realizations of random variables (X,Y)∼p(x,y), where *X* is the continuous input, and its corresponding discrete label is *Y*. Realizations of *X* and *Y* are denoted by $x\in R^{D}$ and y∈{1,…,c}. The codebook $\tilde{X}$ serves as a compressed representation of *X*.

We constructed a probabilistic encoder $p(\tilde{x}|x)$—which is implemented by a deep neural network—that maps an input x to a ***k-way d-dimensional code***
$\tilde{x}\in {\left\{1,2,\ldots ,k\right\}}^{d}$. That is, each entry of $\tilde{x}$ takes on one of *k* possible values, and the cardinality of $\tilde{X}$ is $|\tilde{X}|=k^{d}$. Special cases of this coding scheme include *k*-way class labels (d=1), *d*-dimensional binary codes (k=2), and even fixed-length decimal integers (k=10).

We now describe our model which produces discrete infomax codes. A neural network encoder f(x;θ) outputs *k*-dimensional categorical distributions, $\mathrm{Cat}(p_{i,1},\ldots ,p_{i,k})$. Here, $p_{i,j}\left(x\right)$ represents the probability that output variable *i* takes on value *j*, consuming x as an input, for i=1,…,d and j=1,…,k. The encoder takes x as an input to produce logits $l_{i,j}=f{\left(x\right)}_{i,j}$, which form a d×k matrix:(1)$$\begin{array}{c} \left[\begin{array}{ccccc} l_{1,1} & l_{1,2} & l_{1,3} & \cdots & l_{1,k}\\ l_{2,1} & l_{2,2} & l_{2,3} & \cdots & l_{2,k}\\ \vdots & \vdots & \vdots & \ddots & \vdots \\ l_{d,1} & l_{d,2} & l_{d,3} & \cdots & l_{d,k} \end{array}\right]. \end{array}$$
These logits undergo softmax functions to yield
(2)$$\begin{array}{c} p_{i,j}=\frac{\exp(l_{i,j})}{\sum_{j=1}^{k}\exp\left(l_{i,j}\right)}. \end{array}$$
Each example x in the training set is assigned a codeword $\tilde{x}={\left[{\tilde{x}}_{1},\ldots ,{\tilde{x}}_{d}\right]}^{⊤}$, each entry of which is determined by one of *k* events that is most probable; i.e.,
(3)$$\begin{array}{c} \tilde{x}={\left[\overset{{\tilde{x}}_{1}}{\overbrace{\underset{j}{arg max}p_{1j}}},\;\cdots ,\;\overset{{\tilde{x}}_{d}}{\overbrace{\underset{j}{arg max}p_{dj}}}\right]}^{⊤}. \end{array}$$
While the stochastic encoder $p(\tilde{x}|x)$ induces a soft partitioning of input data, codewords assigned by the rule in (Equation 3) yield a hard partitioning of *X*.

### 2.2. Loss Function

The *i*-th symbol is assumed to be sampled from the resulting categorical distribution $\mathrm{Cat}(p_{i1},\ldots ,p_{ik})$. We denote the resulting distribution over codes as $\tilde{X}$ and a code as $\tilde{x}$. Instead of sampling $\tilde{x}∼\tilde{X}$ during training, we use a loss function that optimizes the expected performance of the entire distribution $\tilde{X}$.

We train the encoder by maximizing the mutual information between the distributions of codes $\tilde{X}$ and labels *Y*. The mutual information is a symmetric quantity that measures the amount of information shared between two random variables. It is defined as
(4)$$\begin{array}{c} I\left(\tilde{X};Y\right)=H\left(\tilde{X}\right)-H\left(\tilde{X}|Y\right). \end{array}$$
Since $\tilde{X}$ and *Y* are discrete, their mutual information is bounded from both above and below as $0\leq I\left(\tilde{X};Y\right)\leq \log\left|\tilde{X}\right|=d\log k$. To optimize the mutual information, the encoder directly computes empirical estimates of the two terms on the right-hand side of (Equation 4). Note that both terms consist of entropies of categorical distributions, which have the general closed-form formula:(5)$$\begin{array}{c} H\left(p_{1},\ldots ,p_{n}\right)=\sum_{i=1}^{n}p_{i}\log p_{i}. \end{array}$$
Let ${\bar{p}}_{ij}$ be the empirical average of $p_{ij}$ calculated using data points in a batch. Then, ${\bar{p}}_{ij}$ is an empirical estimate of the marginal distribution $p(\tilde{x})$. We compute the empirical estimate of $H(\tilde{X})$ by adding its entropy estimate for each dimension.
(6)$$\begin{array}{cc} \hat{H}\left(\tilde{X}\right)\; & =\sum_{i=1}^{d}H\left({\bar{p}}_{i1},\ldots ,{\bar{p}}_{ik}\right). \end{array}$$
We can also compute
(7)$$\begin{array}{c} \hat{H}\left(\tilde{X}|Y\right)=\sum_{y=1}^{c}p\left(Y=y\right)\hat{H}\left(\tilde{X}|Y=y\right), \end{array}$$
where *c* is the number of classes. The marginal probability p(Y=y) is the frequency of class *y* in the minibatch, and $H(\tilde{X}|Y=y)$ can be computed by computing (Equation 6) using only datapoints which belong to class *y*. We emphasize that such a closed-form estimation of $I(\tilde{X};Y)$ is only possible because we are using discrete codes. If $\tilde{X}$ were instead a continuous variable, we would only be able to maximize an approximation of $I(\tilde{X};Y)$ (e.g., Belghazi et al. [7]).

We briefly examine the loss function (Equation 4) to see why maximizing it results in discriminative $\tilde{X}$. Maximizing $H(\tilde{X})$ encourages the distribution of all codes to be as dispersed as possible, and minimizing $H(\tilde{X}|Y)$ encourages the average embedding of each class to be as concentrated as possible. Thus, the overall loss $I(\tilde{X};Y)$ imposes a partitioning problem on the model: it learns to split the entire probability space into regions with minimal overlap between different classes. As this problem is intractable for the large models considered in this work, we seek to find a local minima via stochastic gradient descent (SGD). We provide a further analysis of this loss function in Section 3.1.

### 2.3. Similarity Measure

Suppose that all data points in the training set are assigned their codewords according to the rule (Equation 3). Now we introduce how to compute a similarity between a query datapoint $x^{\left(q\right)}$ and a support datapoint $x^{\left(s\right)}$ for information retrieval or few-shot classification, where the superscripts (q),(s) stand for query and support, respectively. Denote by ${\tilde{x}}^{\left(s\right)}$ the codeword associated with $x^{\left(s\right)}$, constructed by (Equation 3). For the test data $x^{\left(q\right)}$, the encoder yields $p_{i,j}\left(x^{\left(q\right)}\right)$ for i=1,…,d and j=1,…,k. As a similarity measure between $x^{\left(q\right)}$ and $x^{\left(s\right)}$, we calculate the following log probability.
(8)$$\begin{array}{c} \sum_{i=1}^{d}\log p_{i,{\tilde{x}}_{i}^{\left(s\right)}}\left(x^{\left(q\right)}\right). \end{array}$$
The probabilistic quantity (Equation 8) indicates that $x^{\left(q\right)}$ and $x^{\left(s\right)}$ become more similar when the encoder’s output—when $x^{\left(q\right)}$ is provided—is well aligned with ${\tilde{x}}^{\left(s\right)}$.

We can view our similarity measure (Equation 8) as a probabilistic generalization of the Hamming distance [8]. The Hamming distance quantifies the similarity between two strings of equal length as the number of positions at which the corresponding symbols are equal. As we have access to a distribution over codes, we use (Equation 8) to directly compute the log probability of having the same symbol at each position.

We use (Equation 8) as a similarity metric for both few-shot classification and image retrieval. We perform few-shot classification by computing a codeword for each class via (Equation 3) and classifying each test image by choosing the class that has the highest value of (Equation 8). We similarly perform image retrieval by mapping each support image to its most likely code (Equation 3) and for each query image retrieving the support image that has the highest (Equation 8).

While we have described the operations in (Equation 3) and (Equation 8) for a single pair $\left(x^{\left(q\right)},x^{\left(s\right)}\right)$, one can easily parallelize our evaluation procedure, since it is an argmax followed by a sum. Furthermore, $\tilde{x}$ typically requires little memory, as it consists of discrete values, allowing us to compare against large support sets in parallel. Experiments in Section 5.4 investigate the degree of DIMCO’s efficiency in terms of both time and memory.

### 2.4. Regularizing by Enforcing Independence

One way of interpreting the code distribution $\tilde{X}$ is as a group of *d* separate code distributions ${\tilde{x}}_{1},\ldots ,{\tilde{x}}_{d}$. Note that the similarity measure described in (Equation 8) can be seen as ensemble of the similarity measures of these *d* models. A classic result in ensemble learning is that using more diverse learners increases ensemble performance [9]. In a similar spirit, we used an optional regularizer which promotes pairwise independence between each pair in these *d* codes. Using this regularizer stabilized training, especially in more large-scale problems.

Specifically, we randomly sample pairs of indices $i_{1},i_{2}$ from 1,…,d during each forward pass. Note that ${\tilde{x}}_{i_{1}}⊗{\tilde{x}}_{i_{2}}$ and $\left({\tilde{x}}_{i_{1}},{\tilde{x}}_{i_{2}}\right)$ are both categorical distributions with support size $k^{2}$, and that we can estimate the two different distributions within each batch. We minimize their KL divergence to promote independence between these two distributions: (9)$$\begin{array}{c} L_{\mathrm{ind}}=D_{KL}({\tilde{x}}_{i_{1}}⊗{\tilde{x}}_{i_{2}}\|{\tilde{x}}_{i_{1}},{\tilde{x}}_{i_{2}}). \end{array}$$
We compute (Equation 9) for a fixed number of random pairs of indices for each batch. The cost of computing this regularization term is miniscule compared to that of other components such as feeding data through the encoder.

Using this regularizer in conjunction with the learning objective (Equation 4) yields the following regularized loss:(10)$$\begin{array}{c} L=-I\left(\tilde{X};Y\right)+\lambda L_{\mathrm{ind}}=-H\left(\tilde{X}\right)+H\left(\tilde{X}|Y\right)+\lambda L_{\mathrm{ind}}. \end{array}$$
We fix λ=1 in all experiments, as we found that DIMCO’s performance was not particularly sensitive to this hyperparameter. We emphasize that while this optional regularizer stabilizes training, our learning objective is the mutual information $I(\tilde{X};Y)$ in (Equation 4).

### 2.5. Visualization of Codes

In Figure 2, we show images retrieved using our similarity measure (Equation 8). We trained a DIMCO model (k=16, d=4) on the CIFAR100 dataset. We selected specific code locations and plotted the top 10 test images according to our similarity measure. For example, the top (leftmost) image for code $\left(\;\cdot \;,j_{2},\;\cdot \;,j_{4}\right)$ would be computed as
(11)$$\begin{array}{c} \underset{n\in \{1,\ldots ,N\}}{arg max}\left[\log p_{2,j_{2}}\left(x^{n}\right)+\log p_{4,j_{4}}\left(x^{n}\right)\right], \end{array}$$
where *N* is the number of test images.

We visualize two different combinations of codes in Figure 2. The two examples show that using codewords together results in their respective semantic concepts being combined: (man + fish = man holding fish), (round + warm color = orange). While we visualized combinations of 2 codewords for clarity, DIMCO itself uses a combination of *d* such codewords. The regularizer described in Section 2.4 further encourages each of these *d* codewords to represent different concepts. The combinatorially many ($k^{d}$) combinations in which DIMCO can assemble such codewords gives DIMCO sufficient expressive power to solve challenging tasks.

## 3. Analysis

### 3.1. Is Mutual Information a Good Objective?

Our learning objective for DIMCO (Equation 4) is the mutual information between codes and labels. In this subsection, we justify this choice by showing that many previous objectives are closely related to mutual information. Due to space constraints, we only show high-level connections here and provide a more detailed exposition in Appendix A.

#### 3.1.1. Cross-Entropy

The de facto loss for classification is the cross-entropy loss, which is defined as
(12)$$\begin{array}{c} \mathrm{xent}\left(Y,X\right)=-E_{y∼Y,x∼X}\left[\log q(\hat{Y}=y|\tilde{X}\left(x\right))\right], \end{array}$$
where $\hat{Y}$ is the model’s prediction of *Y*. Using the observation that the final layer q(·) acts a parameterized approximation of the true conditional distribution $p(Y|\tilde{X})$, we write this as
(13)$$\begin{array}{ccc} \mathrm{xent}(Y,X) & \approx & E_{y∼Y,x∼X}\left[-\log p(y|\tilde{x})\right]\\ & = & -I\left(\tilde{X};Y\right)+H\left(Y\right). \end{array}$$
The H(Y) term can be ignored since it is not affected by model parameters. Therefore, minimizing cross-entropy is approximately equivalent to maximizing mutual information. The two objectives become completely equivalent when the final linear layer q(·) perfectly represents the conditional distribution $q(y|\tilde{x})$. Note that for discrete $\tilde{x}$, we cannot use a linear layer to parameterize $q(y|\tilde{x})$, and therefore, cannot directly optimize the cross-entropy loss. We can therefore view our loss as a necessary modification of the cross-entropy loss for our setup of using discrete embeddings.

#### 3.1.2. Contrastive Losses

Many metric learning methods [1,2,10,11,12] use a contrastive learning objective to learn a continuous embedding ($\tilde{X}$). Such contrastive losses consist of (1) a positive term that encourages an embedding to move closer to that of other relevant embeddings and (2) a negative term that encourages it to move away from irrelevant embeddings. The positive term approximately minimizes $\log p(\tilde{X}|y)$, and the negative term as approximately minimizes $\log p(\tilde{X})$. Together, these terms have the combined effect of maximizing
(14)$$\begin{array}{c} E\left[\log p\left(\tilde{x}|y\right)-\log p\left(\tilde{x}\right)\right]\\ \;\;\;=-H\left(\tilde{X}|Y\right)+H\left(\tilde{X}\right)=I\left(\tilde{X};Y\right). \end{array}$$
We show such equivalences in detail in Appendix A.

In addition to these direct connections to previous loss functions, we show empirically in Section 5.1 that the mutual information strongly correlates with both the top-1 accuracy metric for classification and the Recall@1 metric for retrieval.

### 3.2. Does Using Discrete Codes Help Generalization?

In Section 1, we have provided motivation for the use of discrete codes through the regularization effect of an information bottleneck. In this subsection, we theoretically analyze whether learning discrete codes by maximizing mutual information leads to better generalization. In particular, we study how the mutual information on the test set is affected by the choice of input dataset structure and code hyperparameters *k* and *d* through a PAC learning bound.

We analyze DIMCO’s characteristics at the level of minibatches. Following related meta-learning works [13,14], we call each batch a “task”. We note that this is only a difference in naming convention, and our analysis applies equally well to the metric learning setup: we can view each batch consisting of support and query points as a task.

Define a task *T* to be a distribution over Z=X×Y. Let tasks $T^{1},\ldots ,T^{n}$ be sampled i.i.d. from a distribution of tasks τ. Each task *T* consists of a fixed-size dataset $D_{T}=z_{T}^{1},\ldots ,z_{T}^{m}=\left(x_{T}^{1},y_{T}^{1}\right),\ldots ,\left(x_{T}^{m},y_{T}^{m}\right)$, which is a set of *m* i.i.d. samples from the data distribution ($z_{T}^{j}∼T$). Let θ be the parameters of DIMCO. Let $X,Y,\tilde{X}$ be the random variables for data, labels, and codes, respectively. Recall that our objective is the expected mutual information between labels and codes:(15)$$\begin{array}{c} L\left(\tau ,\theta \right)=-E_{T∼\tau }\left[I(\tilde{X}\left(X_{T},\theta \right);Y_{T})\right]. \end{array}$$
The loss that we actually optimize (Equations (Equation 6) and (Equation 7)) is the empirical loss:(16)$$\begin{array}{c} \hat{L}\left(T^{1:n},\theta \right)=-\frac{1}{n}\sum_{i=1}^{n}\hat{I}\left(\tilde{X}\left(X_{T^{i}},\theta \right);Y_{T^{i}}\right). \end{array}$$

The following theorem bounds the difference between the expected loss L and the empirical loss $\hat{L}$.

**Theorem** **1.**
*Let $d_{\Theta }$ be the VC dimension of the encoder $\tilde{X}\left(\;\cdot \;\right)$. The following inequality holds with high probability:*

(17)
$$\begin{array}{c} L\left(\tau ,\theta \right)-\hat{L}\left(T^{1:n},\theta \right)\leq O\left(\sqrt{\frac{d_{\Theta }}{n}\log\frac{n}{d_{\Theta }}}\right)+O\left(\frac{|\tilde{X}|\log\left(m\right)}{\sqrt{m}}\right)+O\left(\frac{|\tilde{X}\left||Y\right|}{m}\right). \end{array}$$



**Proof.** We use VC dimension bounds and a finite sample bound for mutual information [15]. We defer a detailed statement and proof to Appendix B.    □

First note that all three terms in our generalization gap (Equation 29) converge to zero as n,m→∞. This shows that training a model by maximizing empirical mutual information, as in Equations (Equation 6) and (Equation 7), generalizes perfectly in the limit of infinite data.

Theorem 1 also shows how the generalization gap is affected differently by dataset size *m* and number of datasets *n*. A large *n* directly compensates for using a large backbone ($d_{\Theta }$), and a large *m* compensates for using a large final representation ($\tilde{X}$). Put differently, to effectively learn from small datasets (*m*), one should use a small representation $\left(\tilde{X}\right)$. The number of datasets *n* is typically less of a problem because the number of different ways to sample datasets is combinatorially large (e.g., $n>10^{10}$ for miniImagenet 5-way 1-shot tasks). Recall that DIMCO has $|\tilde{X}|=d\log k$, meaning that we can control the latter two terms using our hyperparameters d,k. We have explained the use of discrete codes through the information bottleneck effect of small codes $\tilde{X}$, and Theorem 1 confirms this intuition.

## 4. Related Work

**Information bottleneck.** DIMCO and Theorem 1 are both close in spirit to the information bottleneck (IB) principle [5,6,16]. IB finds a set of compact representatives $\tilde{X}$ while maintaining sufficient information about *Y*, minimizing the following objective function:(18)$$\begin{array}{c} J\left(p\left(\tilde{x}|x\right)\right)=I\left(\tilde{X};X\right)-\beta I\left(\tilde{X};Y\right), \end{array}$$
subject to $\sum_{\tilde{x}}p\left(\tilde{x}|x\right)=1$. Equivalently, it can be stated that one maximizes $I(\tilde{X};Y)$ while simultaneously minimizing $I(\tilde{X};X)$. Similarly, our objective (Equation 15) is information maximization $I(\tilde{X};Y)$, and our bound (Equation 29) suggests that the representation capacity $|\tilde{X}|$ should be low for generalization. In the deterministic information bottleneck [17], $I(\tilde{X};X)$ is replaced by $H(\tilde{X})$. These three approaches to generalization are related via the chain of inequalities $I\left(\tilde{X};X\right)\leq H\left(\tilde{X}\right)\leq \log\left|\tilde{X}\right|$, which is tight in the limit of $\tilde{X}$ being imcompressible. For any finite representation, i.e., $|\tilde{X}|=N$, the limit β→∞ in (Equation 18) yields a hard partitioning of *X* into *N* disjoint sets. DIMCO uses the infomax principle to learn $N=k^{d}$ such representatives, which are arranged by *k*-way *d*-dimensional discrete codes for compact representation with sufficient information on *Y*.

**Regularizing meta-learning.** Previous meta-learning methods have restricted task-specific learning by learning only a subset of the network [18], learning on a low-dimensional latent space [19], learning on a meta-learned prior distribution of parameters [20], and learning context vectors instead of model parameters [21]. Our analysis in Theorem 1 suggests that reducing the expressive power of the task-specific learner has a meta-regularizing effect, indirectly giving theoretical support for previous works that benefited from reducing the expressive power of task-specific learners.

**Discrete representations.** Discrete representations have been thoroughly studied in information theory [22]. Recent deep learning methods directly learn discrete representations by learning generative models with discrete latent variables [23,24,25] or maximizing the mutual information between representation and data [26]. DIMCO is related to but differs from these works, as it assumes a supervised meta-learning setting and performs infomax using *labels* instead of data.

A standard approach to learning label-aware discrete codes is to first learn continuous embeddings and then quantize it using an objective that maximally preserves its information [27,28,29]. DIMCO can be seen as an end-to-end alternative to quantization which directly learns discrete codes. Jeong and Song [30] similarly learns a sparse binary code in an end-to-end fashion by solving a minimum cost flow problem with respect to labels. Their method differs from DIMCO, which learns a dense discrete code by optimizing $I(\tilde{X};Y)$, which we estimate with a closed-form formula.

**Metric learning.** The structure and loss function of DIMCO are closely related to those of metric learning methods [1,11,12,31]. We show that the loss functions of these methods can be seen as approximations of the mutual information ($I(\tilde{X};Y)$) in Section 2.2, and provide more in-depth exposition in Appendix A. While all of these previous methods require a support/query split within each batch, DIMCO simply optimizes an information-theoretic quantity of each batch, removing the need for such structured batch construction.

**Information theory and representation learning.** Many works have applied information-theoretic principles to unsupervised representation learning: to derive an objective for GANs to learn disentangled features [32], to analyze the evidence lower bound (ELBO) [33,34], and to directly learn representations [35,36,37,38,39]. Related also are previous methods that enforce independence within an embedding [40,41]. DIMCO is also an information-theoretic representation learning method, but we instead assume a supervised learning setup where the representation must reflect ground-truth labels. We also used previous results from information theory to prove a generalization bound for our representation learning method.

## 5. Experiments

In our experiments, we used datasets with varying degrees of complexity: CIFAR10/100 [42], miniImageNet [31], CUB200 [43], Cars196 [44], and ImageNet (ILSVRC-2012-CLS, Deng et al. [45]). We used standard train/test splits for each dataset unless stated otherwise. We also used various network architectures: 4-layer convnet [31] and ResNet12/20/50 [46,47]. We followed previously reported experimental setups as closely as possible, and provide minor experiment details in Appendix C.

### 5.1. Correlation of Metrics

We have shown in Section 3.1 that the mutual information $I(\tilde{X};Y)$ is strongly connected to previous loss functions for classification and retrieval. In this subsection, we show experiments performed to verify whether $I(\tilde{X};Y)$ is a good *metric* that quantitatively shows the quality of the representation $\tilde{X}$. We trained DIMCO on the miniImageNet dataset with k=d=64 for 20 epochs. We plot the pairwise correlations between five different metrics: (5,10,20)-way 1-shot accuracy, Recall@1, and $I(\tilde{X};Y)$. The results in Figure 3 show that all five metrics are very strongly correlated. We observed similar trends when training with loss functions other than $I(\tilde{X};Y)$ as well; we show these experiments in Appendix C due to space constraints.

### 5.2. Label-Aware Compression

We applied DIMCO to compressing feature vectors of trained classifier networks. We obtained penultimate embeddings of ResNet20 networks each trained on CIFAR10 and CIFAR100. The two networks had top-1 accuracies of 91.65 and 66.61, respectively. We trained on embeddings for the train set of each dataset, and measured top-1 accuracy of the test set using the training set as support. We compare DIMCO to product quantization (PQ, Jegou et al. [28]), which similarly compresses a given embededing to a *k*-way *d*-dimensional code. We compare the two methods in Table 1 with the same range of k,d hyperparameters. We performed the same experiment on the larger ImageNet dataset with a ResNet50 network which had a top-1 accuracy of 76.00. We compare DIMCO to both adaptive scalar quantization (SQ) and PQ in Table 2. We show extended experiments for all three datasets in Appendix A.

The results in Table 1 and Table 2 demonstrate that DIMCO consistently outperforms PQ, and is especially efficient when *d* is low. Furthermore, the ImageNet experiment (Table 2) shows that DIMCO even outperforms SQ, which has a much lower compression rate compared to the embedding sizes we consider for DIMCO. These results are likely due to DIMCO performing *label-aware compression*, where it compresses the embedding while taking the label into account, whereas PQ and SQ only compress the embeddings themselves.

**Figure 3 entropy-24-00501-f003:**
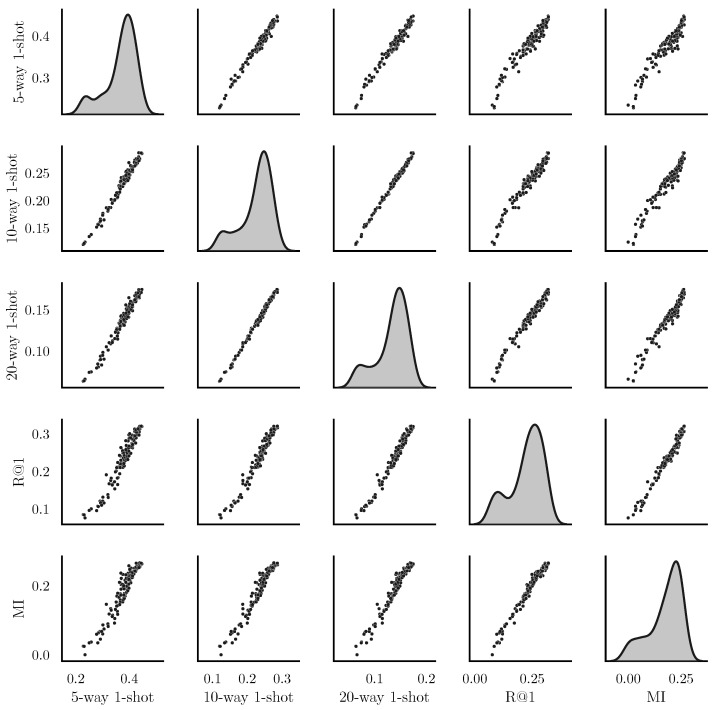
Pairwise correlations between $\mathrm{MI}=I(X;\tilde{X})$ and previous metrics. Best viewed zoomed in.

### 5.3. Few-Shot Classification

We evaluated DIMCO’s few-shot classification performance on the miniImageNet dataset. We compare our method against the following previous works: Snell et al. [3], Vinyals et al. [31], Liu et al. [48], Ye et al. [49], Ravi and Larochelle [50], Sung et al. [51], Bertinetto et al. [52], Lee et al. [53]. All methods use the standard four-layer convnet with 64 filters per layer. While some methods used more filters, we used 64 for a fair comparison. We used the data augmentation scheme proposed by Lee et al. [53] and used balanced batches of 100 images consisting of 10 different classes. We evaluated both 5-way 1-shot and 5-way 5-shot learning, and report 95% confidence intervals of 1000 random episodes on the test split.

Results are shown in Table 3, and we provide an extended table with an alternative backbone in Appendix A. Figure Table 3 shows that DIMCO outperforms previous works on the 5-way 1-shot benchmark. DIMCO’s 5-way 5-shot performance is relatively low, likely because the similarity metric (Section 2.3) handles support datapoints individually instead of aggregating them, similarly to Matching Nets [31]. Additionally, other methods are explicitly trained to optimize 5-shot performance, whereas DIMCO’s training procedure is the same regardless of task structure.

### 5.4. Image Retrieval

We conducted image retrieval experiments using two standard benchmark datasets: CUB-200-2011 and Cars-196. As baselines, we used three widely adopted metric learning methods: Binomial Deviance [54], Triplet loss [1], and Proxy-NCA [2]. The backbone for all methods was a ResNet-50 network pretrained on the ImageNet dataset. We trained DIMCO on various combinations of (p,d), and set the embedding dimension of the baseline methods to 128. We measured the time per query for each method on a Xeon E5-2650 CPU without any parallelization. We note that computing the retrieval time using a parallel implementation would skew the results even more in favor of DIMCO, since DIMCO’s evaluation is simply one memory access followed by a sum.

Results presented in Table 4 show that DIMCO outperforms all three baseline, and that the compact code of DIMCO takes roughly an order of magnitude less memory, and requires less query time as well. This experiment also demonstrates that discrete representations can outperform modern methods that use continuous embeddings, even on this relatively large-scale task. Additionally, this experiment shows that DIMCO can train using large backbones without significantly overfitting.

## 6. Discussion

We introduced DIMCO, a model that learns a discrete representation of data by directly optimizing the mutual information with the label. To evaluate our initial intuition that shorter representations generalize better between tasks, we provided generalization bounds that get tighter as the representation gets shorter. Our experiments demonstrated that DIMCO is effective at both compressing a continuous embedding, and also at learning a discrete embedding from scratch in an end-to-end manner. The discrete embeddings of DIMCO outperformed recent continuous feature extraction methods while also being more efficient in terms of both memory and time. We believe the tradeoff between discrete and continuous embeddings is an exciting area for future research.

DIMCO was motivated by concepts such as the minimum description length (MDL) principle and the information bottleneck: compact task representations should have less room to overfit. Interestingly, Yin et al. [55] reports that doing the opposite—regularizing the task-general parameters—prevents meta-overfitting by discouraging the meta-learning model from memorizing the given set of tasks. In future work, we will investigate the common principle underlying these seemingly contradictory approaches for a fuller understanding of meta-generalization.

## Figures and Tables

**Figure 1 entropy-24-00501-f001:**
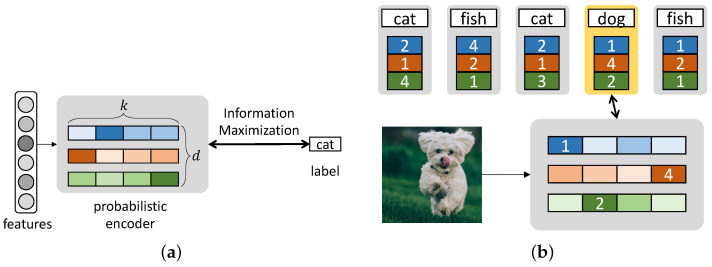
A graphical overview of discrete infomax codes (DIMCO). (**a**) Discrete codes are produced by a probabilistic encoder that maps each datapoint to a distribution over *k*-way *d*-dimensional discrete codes. The encoder is trained to maximize the mutual information between code distribution and label distribution. (**b**) Given a query image, we compare it against a support set of discrete codes and corresponding labels. Our similarity metric is the query’s log probability for each discrete code.

**Figure 2 entropy-24-00501-f002:**
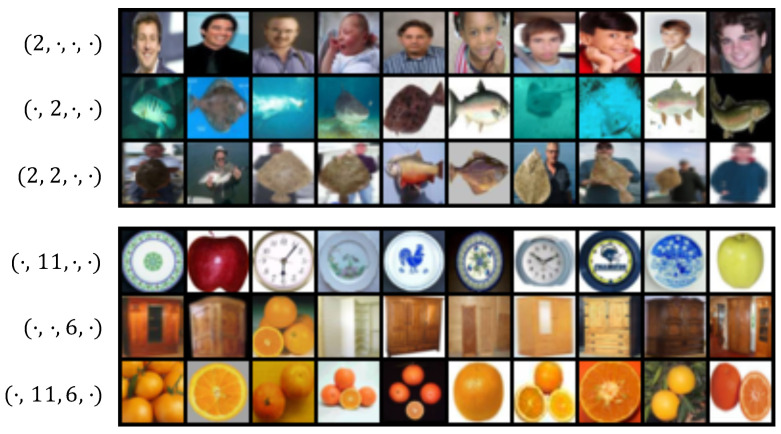
Compositionality of codes in a DIMCO model (k=16, d=4) trained on the CIFAR-100 dataset. Each row shows the top 10 images that assign highest marginal probability to specific codes (shown on left). Composed codewords retrieve images with combined semantic meaning, such as man + fish = man holding fish. We additionally visualize all kd codewords in Appendix A.

**Table 1 entropy-24-00501-t001:** Top-1 accuracies under various compressed CIFAR-10/100 embeddings with size (k,d), with best results for each setting in bold. The original embeddings were continuous 64-dimensional embeddings ($k\approx 2^{32},d=64$).

		CIFAR-10		CIFAR-100	
		*d* = 4	*d* = 16	*d* = 4	*d* = 16
PQ	k=2	50.38	87.24	7.77	32.47
	k=4	86.86	90.92	18.78	52.02
	k=8	90.44	91.49	31.04	58.82
	k=16	91.02	91.52	43.16	61.00
DIMCO	k=2	64.43	88.85	10.03	33.46
(ours)	k=4	88.47	91.04	25.7	53.36
	k=8	91.29	91.45	46.22	58.84
	k=16	**91.68**	**91.57**	**57.83**	**62.49**

**Table 2 entropy-24-00501-t002:** Top-1 accuracies and compression rates of ImageNet embeddings under various compressed embedding sizes (k,d), with best results in bold. The compression rate is the ratio between uncompressed and compressed sizes; it is calculated as $\frac{32\;\cdot \;2048}{d\log k}$.

Method	(*k*,*d*)	Compression Rate	Accuracy
SQ	(2,2048)	32	0.10
SQ	(4,2048)	16	0.28
SQ	(8,2048)	10	12.36
SQ	(16,2048)	8	57.80
PQ	(2,2)	32,768	0.21
PQ	(4,4)	8192	0.92
PQ	(8,8)	2730	11.91
PQ	(16,16)	1024	44.93
DIMCO	(2,2)	32,768	0.14
DIMCO	(4,4)	8192	2.68
DIMCO	(8,8)	2730	33.28
DIMCO	(16,16)	1024	**63.64**

**Table 3 entropy-24-00501-t003:** Few-shot classification accuracies on the miniImageNet benchmark, with best results for each setting in bold. ^†^ denotes transductive methods, which are more expressive by taking unlabeled examples into account.

Method	5-Way 1-Shot	5-Way 5-Shot
${}^{†}$ TPN	55.51 ± 0.86	69.86 ± 0.65
${}^{†}$ FEAT	55.75 ± 0.20	72.17 ± 0.16
MetaLSTM	43.44 ± 0.77	60.60 ± 0.71
MatchingNet	43.56 ± 0.84	55.31 ± 0.73
ProtoNet	49.42 ± 0.78	68.20 ± 0.66
RelationNet	50.44 ± 0.82	65.32 ± 0.70
R2D2	51.2 ± 0.6	**68.8 ± 0.1**
MetaOptNet-SVM	52.87 ± 0.57	68.76 ± 0.48
DIMCO (64,64)	47.33 ± 0.46	61.59 ± 0.52
DIMCO (64,128)	**53.29 ± 0.47**	64.79 ± 0.57

**Table 4 entropy-24-00501-t004:** Image retrieval performance on CUB-200-2011 and Cars-196, measured by Recall@1, with best results for each setting in bold. Memory is the number of bits that an embedding vector of each image uses. Time is seconds taken to retrieve a single query from database (5924 and 8131 images for CUB-200-2011 and Cars-196, respectively).

			CUB-200-2011		Cars-196	
Method	(*k*,*d*)	Memory [Bits]	Recall@1	Time [s]	Recall@1	Time [s]
Binomial Deviance	(-, 128)	4096	57.25	16.37	72.53	21.86
Triplet	(-, 128)	4096	56.80	16.37	73.79	21.86
Proxy-NCA	(-, 128)	4096	56.19	16.37	75.94	21.86
DIMCO (ours)	(32, 32)	160	51.04	1.48	63.44	2.64
	(64, 64)	384	55.78	2.85	72.06	6.01
	(128, 128)	896	58.05	5.81	76.04	11.92
	(256, 256)	2048	**58.90**	12.20	**77.32**	16.04

## Data Availability

Not applicable.

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
