# Peer review of "Discrete Infomax Codes for Supervised Representation Learning"

_entropy, 2022, doi:10.3390/e24040501_

Round 1
Reviewer 1 Report
The manuscript made a favorable impression on me. First, the problem of data compression, which takes into account known class labels, is important in practical applications of pattern recognition. Secondly, the algorithm for obtaining k-way d-dimensional codes associated with input data, developed by the authors, contains elements of novelty, although it uses traditional ideas and methods of supervised representation learning. Thirdly, the computational experiments carried out by the authors illustrate the advantages of the proposed algorithm in comparison with other known methods on adequately selected test data sets.
However, I would like to make a few small comments that may help the authors improve the text of the work.
p.2, line 45 – please finish the sentence.
p.2, line 55 – Are the random variables continuous or discrete?
- 3, line 67 – Please insert a formula for finding logits $l_{I,j}$
- 3, line 74 – The $i$-th symbol (of what?)
- 4, line 85 – what is the complexity of maximization problem (is it NP-hard)?
- 4, line 97 – Is it correct that (q) stands for “query”, while (s) means “support” datapoints? Please check the sentence and the next paragraph.
- 4, line 98 – I do not find this ref. satisfactory.
- 12, line 336 – What is the connection between your problem and the problem of feature extraction?
Overall, I am confused by the abundance of arXiv preprints in the references. Perhaps, they are good and important works, but the lack of peer review that occurs when papers are published in peer-reviewed journals raises some doubts about the reliability of the results presented in them.
Perhaps, in Abstract (or at the beginning of Introduction), it would be good to give a broader context of the problem under consideration in order to interest a wider readership.
Author Response
We thank the reviewer for their helpful comments.
> Give a broader context of the problem under consideration
Thank you for this suggestion, we have updated the abstract to give a broader context for why a compact representation is useful in the problems we consider.
> Abundance of arXiv preprints
Most of the papers we cited are peer-reviewed articles, and we were using the bibtex for the preprint versions. We have updated the citations to reflect the peer-reviewed versions.
> Other comments
Thank you for the valuable suggestions. We have incorporated these edits, and we believe they have improved the quality of the paper.
Please see attached the revised paper.

Reviewer 2 Report
A question for further consideration: is perhaps your proposal such an advance because in discretising the observations it makes mutual information so much easier to work with?
A couple of minor typos slipped by. Check them out in the attached pdf.

Author Response
We thank the reviewer for their helpful comments.
> is perhaps your proposal such an advance because in discretising the observations it makes mutual information so much easier to work with?
Exactly, we believe that our specific parameterization of discrete codes is in itself a contribution because it makes mutual information easy to optimize exactly. This does come with the tradeoff of giving up the high capacity of continuous variables, which have been shown to excel in certain applications. We hope that future works will explore the limits of maximizing MI using our parameterization.
> minor typos
Thank you for the suggestions, we have incorporated these edits into the paper, in the attachment.

Reviewer 3 Report
This work develops a novel method, discrete infomax codes, for representation learning. In general, the paper is well-written and studies an important and interesting problem. The analysis is simple yet solid to provide some theoretical justifications of the proposed method. I only have some minor comments.
- What is the effect of d and k? If either of them is much larger than c, the number of true labels, are we still able to find d independent or sufficiently separated distributions?
- It would be interesting to add some relevant work in Section 2.4, discussing alternative approaches in the literature about promoting independence between distributions.
Author Response
We thank the reviewer for their helpful comments.
> What is the effect of d and k? If either of them is much larger than c, the number of true labels, are we still able to find d independent or sufficiently separated distributions?
d and k determine the capacity of the codebook, which can express d^k different combinations. Our intuition is that if the capacity is set to be much larger than that of the relationship between X and Y (as is common in continuous embeddings), different dimensions will correspond to the same features, violating our pairwise independence criteria. While such redundant codes have been shown to be useful in other applications, this paper shows that more compact codes are beneficial in the low-data regime.
> It would be interesting to add some relevant work in Section 2.4, discussing alternative approaches in the literature about promoting independence between distributions.
Thank you for this suggestion, we have additionally cited “Disentangling by Factorising” and “Information Dropout: Learning Optimal Representations Through Noisy Computation” in the related work section.